

# Effect of induced hyperopia on fall risk and Fourier transformation of postural sway

Byeong-Yeon Moon, Jae Hyeok Choi, Dong-Sik Yu and Sang-Yeob Kim

Department of Optometry, College of Health Science, Kangwon National University, Samcheok, South Korea

## ABSTRACT

**Background and Purpose:** Fall accidents are a social challenge in Korea and elsewhere. Most previous studies have focused on the effects of reduced visual acuity due to myopia on falls and body balance. The objective of this study was to investigate whether uncorrected hyperopia was a major risk factor for falls and to establish whether the risk of falls was absolutely correlated with visual acuity.

**Methods:** Fifty-one young subjects with a mean age of 22.75 ± 2.13 years were enrolled in this study. To induce hyperopic and myopic refractive errors, spherical lenses of ±1.0–6.0 D (1.0 D stepwise) were used. Under each induced condition, fall risk index and sway power were assessed via Fourier transformation of postural sway using a TETRAX system.

**Results:** The fall risk index for eyes-closed was significantly greater than that of eyes-open with full correction ($t = -5.876$, $p < 0.05$). The fall risk index increased significantly from hyperopia induced with −4.0 D lenses (with visual acuity of 0.69 ± 0.32) compared to eyes-open with full correction ($F = 3.213$, $p < 0.05$). However, there was no significant change in the induced myopia conditions, despite a drastic decline in decimal visual acuity. Sway power increased significantly in the low-to-medium frequency band derived from the peripheral vestibular system when hyperopia was induced. A significant difference was detected in hyperopia induced with −6.0 D lenses compared to eyes-open with full correction ($F = 4.981$, $p = 0.017$).

**Conclusion:** An uncorrected hyperopia rather than myopia may increase the risk of falls, although eyes may show normal visual acuity due to the inherent accommodation mechanism. Our findings suggest that the corrected state of refractive errors is more important than the level of visual acuity as the criteria for appropriate visual input, which contributes to stable posture. Therefore, clinicians should consider the refractive condition, especially the characteristics of hyperopia, when analyzing body balance, and appropriate correction of uncorrected hyperopia to prevent falls.

Corresponding author
Sang-Yeob Kim, syk@kangwon.ac.kr

## INTRODUCTION

Globally, 30% of elderly population aged 65 or older have experienced falls, and recurrent falls have been reported in 50% of them (*Ruchinskas, 2003*). In Korea, too, it is known that 17.2% of the elderly over 65 years of age living in the community experience falls at
least once a year (*Huang et al., 2003*). These frequent falls may result in impaired physical function limiting their daily activities and physical mobility. Also, hospitalization for treatment of fall injuries increases the medical costs of the elderly and the socioeconomic burden associated with poor quality of life (*Kannus et al., 2005*).

Risk factors for falls include visual impairment, weak leg strength, impaired sense of balance and ability to walk, cognitive impairment (*Lawlor, Patel & Ebrahim, 2003*) and chronic disease and drug use (*Jack et al., 1995*). With regard to falls and visual function, *Ivers et al. (1998)* and *Klein et al. (2003)* found that poor visual acuity and low contrast sensitivity were closely related to falls. *Lord & Dayhew (2001)* reported the importance of depth perception to prevent falls. In addition, a previous study reported that falls were associated with poor peripheral vision, which increases the risk of falls by 8% with every 10% reduction in visual field (*Freeman et al., 2007*). Thus, visual acuity, contrast sensitivity, depth perception, and peripheral vision are key visual functions necessary to maintain physical balance. Therefore, visual challenges disturb the physical balance and increase the risk of falls. Visual acuity is the most basic parameter for evaluation of visual function and is closely related to refractive error. Myopia is a typical refractive error that reduces visual acuity. Most previous studies have focused on the effects of reduced visual acuity due to myopia on falls and body balance. *Edwards (1946)* found that induced myopia of +5.0 D in young adults increased postural sway by 51%. *Straube, Paulus & Brandt (1990)* also reported that myopic blurring induced with +4.0 to +6.0 D caused increase in postural sway by 25%, which was increased by 50% with +8.0 D, and by 100% with +10.0 D. Refractive errors may involve hyperopia as well as myopia. In Korea, hyperopia has been reported in approximately 6% of the 7,695 people aged 10–80 years (*Lee & Kwen, 2012*). However, the effects of hyperopic refractive error on the risk of falls have been rarely studied. Hyperopia is characterized by an accommodation mechanism in the eye that favors image formation on the fovea (*Benjamin, 2006*) which results in adequate visual acuity without optical correction. This phenomenon can prevent clear differentiation of refractive error based on a general assessment of visual acuity. Therefore, the study of hyperopia is rare compared with myopia, which is associated with a significant reduction in visual acuity. This study was prompted by our hypothesis that hyperopia is a type of refractive error, which is not similar to visual input from emmetropia, despite appropriate visual acuity resulting from the accommodation mechanism.

Therefore, we investigated the effect of hyperopia and myopia induced by (±) spherical ophthalmic lens on the risk of falls using the TETRAX biofeedback system, in healthy young adults. Furthermore, when the risk of falls is increased by each refractive error, we identified the cause using the Fourier transformation analysis of postural sway, a unique method provided by the TETRAX system.

## MATERIALS AND METHODS

### Subject

The study included 51 healthy young adults (32 males and 19 females) with an average age of 22.75 ± 2.13 years without any challenges associated with physical balance. The mean refractive power of the subjects was S −2.86 ± 2.80 D, C −1.09 ± 1.17 D, and

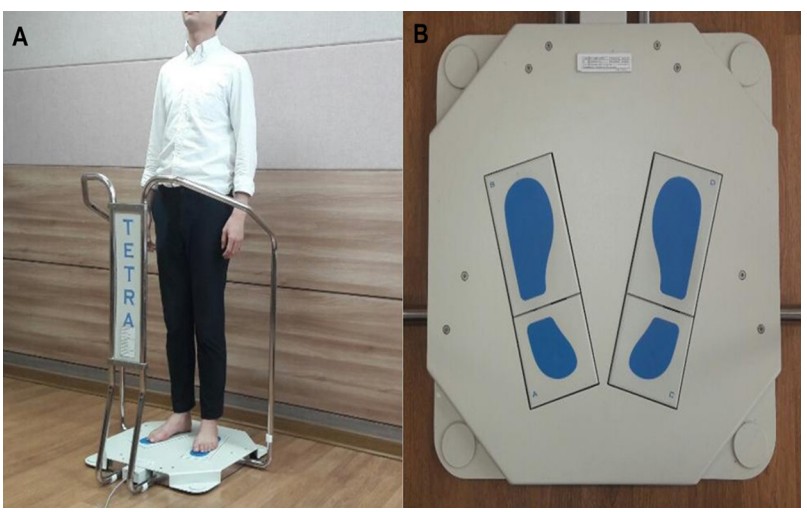

**Figure 1 Instrument for postural assessment used in this study.** (A) Tetrax static posturography device. (B) Four plates on Tetrax device.

the monocular corrected decimal visual acuity was a minimum of 0.9. All subjects had no a history of neuromusculoskeletal diseases, orthopedic disorders, disorders of the vestibular system, strabismus and ophthalmologic diseases, or related drug treatment, which affected body balance. Subjects who manifested signs or symptoms associated with non-strabismic binocular dysfunction or accommodative anomaly were excluded from the study. This clinical study was approved by Kangwon National University's Institutional Review Board (KWNUIRB 2018-04-004-007). We explained the purpose of this study to all participants and received their informed consent form, and conducted all procedures pursuant to the ethical principles of the Helsinki Declaration.

## Analytical equipment

We used the Tetrax biofeedback system (Tetrax Protable Multiple System; Tetrax Ltd., Ranmat Gan, Israel) for measurement and assessment (Fig. 1). The device is equipped with four ground reaction force sensors, labeled A (left foot heel), B (left fore foot), C (right foot heel), and D (right fore foot) for calculation of lateral and back-and-forth movements of one foot, and lateral, back-and-forth, and diagonal movements of both feet to enable highly accurate analysis (*Park & Kang, 2011*; *Chang & Woo, 2010*). The information output from the A–D ground reaction force sensors was converted into a digital signal to analyze the range and velocity of the postural sway and the movement pattern of the center of gravity to facilitate the assessment of variant postures including general stability, fall risk index, and sway power via Fourier transformation. The following parameters are related to the measurement.

*Fall risk index* (*Steinberg et al., 2013*): This index represents the degree of risk of falls, which cause injuries to the body due to sudden shift in body position such as slips or falls from top to bottom. The index is measured by calculating the number of standard deviations from the standard database average provided by the IBS software. The higher

the value of the measured index, the higher is the risk of falls. It is classified into low risk (0–35), moderate risk (36–57), and high risk (58–100) categories.

*Fourier analysis of sway*: Fourier analysis is a mathematical expression of the wave signal of body vibration in the horizontal plane made by the patient to maintain the upright posture (*Kohen-Raz, 1991*). Various frequency components included in the measured value vary when the body sway occurs on the ground in response to reaction force of the device. They are divided into four frequency bands through Fourier transformation to calculate the sway power. It represents the postural sway power at each frequency band and increase in this value suggests instability of the relevant sensory organ responsible for balance. The range of the low frequency band is between 0.01 and 0.10 Hz and an abnormally high value is associated with visual impairment. Low-to-medium frequency band is in the range of 0.10–0.50 Hz, and an abnormally high value is associated with disorders of the peripheral vestibular system. A High-to-medium frequency band is in the 0.50–1.00 Hz range, and an abnormally high value is related to somatosensory abnormality. A high frequency band is in the 1.00–3.00 Hz range, and an abnormally high value is associated with central nervous system disorder (*Sunlight, 2006*). Fourier spectral analysis of postural sway is considered a valuable tool in clinical diagnosis, and Tetrax posturography has demonstrated high test-retest reliability (*Kohen-Raz, 1991*).

## Measurement process

A fully corrected value of the subject's refractive errors was obtained through a subjective refraction test using a Phoropter (Ultramatic RX Master; Reichert, Depew, NY, USA). The examiner asked the subjects to wear a trial frame (Trial frame TF-3; Topcon, Japan) with full correction (optically full-corrected condition). The subject's feet were accurately aligned on the four force plates with the anatomical position and the fall risk index and sway power were recorded after measurement for 32 s according to the measurement manual.

After completing the measurement at the eyes-open with full correction and eyes closed conditions, using a spherical lens of ±1.0 D to ±6.0 D (1.0 D stepwise) in front of both eyes, we induced binocular myopic (induced with + spherical lens) and hyperopic (induced with − spherical lens) refractive error for each degree and randomly measured each type of refractive error compared to the measured values in eyes-open with full correction and eyes-closed conditions. During the measurement, the subjects were instructed to keep looking at the fixed target on the LCD visual acuity test chart (LUCIDLC; Everview, Seoul, Korea) located 6 m ahead. To remove perception changes of magnification changes in the peripheral visual field caused by added lenses, we used a fixation target with a white background. The binocular decimal visual acuity in each refractive error state induced was measured and recorded. A 1 min break was provided every time the lens power was changed and a 10 min break was given when the type of refractive error was changed.

For data analysis, a paired *t*-test was used to compare the mean fall risk index between the eyes-open with full correction and eyes-closed conditions and repeated-measures analysis of variance (ANOVA) was performed using IBM SPSS Statistics 23 to analyze the

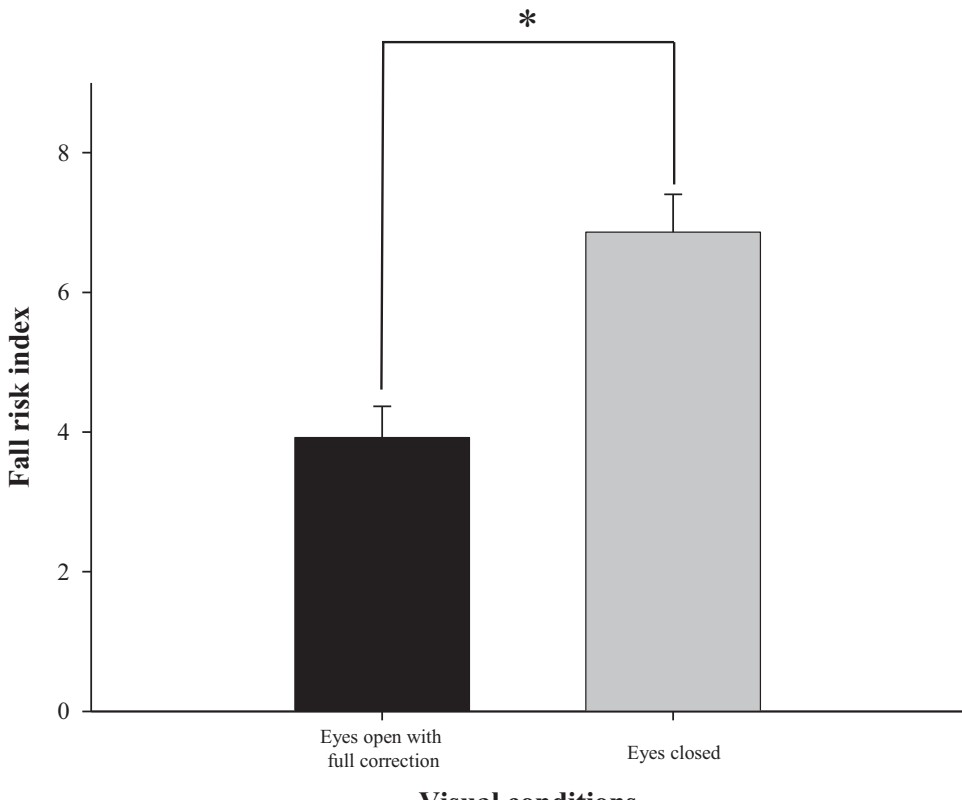

**Figure 2 Comparison of fall risk index between eyes-open with full correction and eyes-closed.**
*Significantly different between eyes-open with full correction and eyes-closed conditions according to paired-$t$ test. Error bars indicate the standard error (SE) of the mean.

changes in the fall risk index and sway power according to each refractive error. A statistically significant difference was observed at $p < 0.05$.

## RESULTS

The difference in fall risk index between eyes-open with full correction and eyes-closed is shown in Fig. 2. When the eyes were closed, the fall risk index increased significantly ($t = -5.876$, $p < 0.05$), which means the risk of fall was increased when the visual information was blocked. Fig. 3 shows changes in binocular decimal visual acuity with the degree of experimentally induced refractive error using a (±) spherical lens. Although visual acuity was decreased with increasing refractive error compared with full correction, even when the same refractive error was induced, a relatively higher visual acuity was maintained with hyperopia compared with myopia. The change in fall risk index with the power of refractive error induced by each type is shown in Fig. 4. When the myopic refractive error increased, the fall risk index also increased but it was not statistically significant. However, when the hyperopic refractive error increased, the fall risk index increased significantly. A post-hoc analysis (least significant difference; LSD) showed that in the hyperopia induced to −4.0 D or higher, the difference from the values measured in

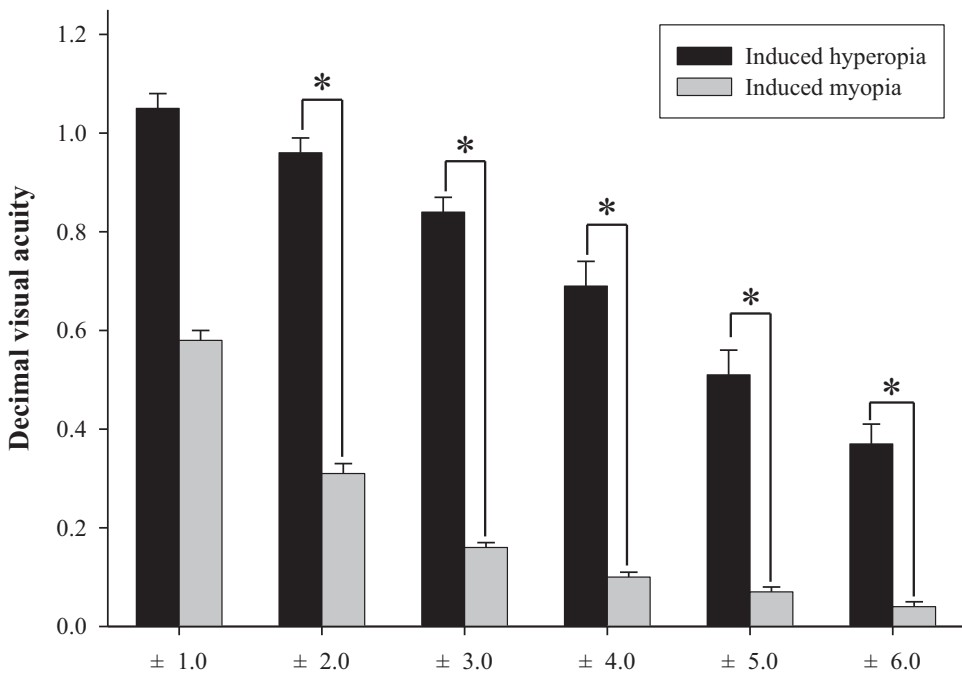

**Figure 3 Comparison of binocular decimal visual acuity between hyperopic and myopic refractive errors induced by (±) spherical lenses.** *Significantly different in visual acuity between induced hyperopia and myopia under similar lens power according to paired-*t* test. Error bars indicate the standard error (SE) of the mean.

condition of eyes-open with full correction was statistically significant ($F = 3.213$, $p < 0.05$, post hoc $p = 0.001$ for −4.0 D, $p = 0.002$ for −5.0 D, $p = 0.001$ for −6.0 D). As shown in Fig. 5, the risk of falls was similar to the risk with the eyes-closed when hyperopia was induced to −3.0 D or higher ($F = 2.989$, $p = 0.015$, post hoc $p = 0.002$ for −1.0 D, $p = 0.030$ for −2.0 D) and when myopia was induced to +6.0 D ($F = 3.397$, $p = 0.008$, post hoc $p = 0.000$ for +1.0 D and +2.0 D, $p = 0.031$ for +3.0 D, $p = 0.023$ for +4.0 D, $p = 0.015$ for +5.0 D). Tables 1 and 2 present the analysis of changes in sway power at each frequency band derived from the relevant sensory organs under each type of refractive error. As shown in Table 1, no significant change in sway power at each frequency band was found even if the myopic refractive error increased. However, in the case of hyperopic refractive error (Table 2), with increase in induced refractive error, the sway power in the low-to-middle frequency band increased proportionately. There was a significant difference in case of hyperopia induced to −6.0 D compared to eyes-open with full correction condition ($F = 4.981$, $p = 0.017$, post hoc $p = 0.003$).

## DISCUSSION

The visual system plays an important role in stabilizing the balance by continuously updating the nervous system with positional information and body movement (*Lord, 2006*). The most traditional experiment demonstrating the role of visual information in physical balance entailed comparison with eyes-open and closed. Several studies

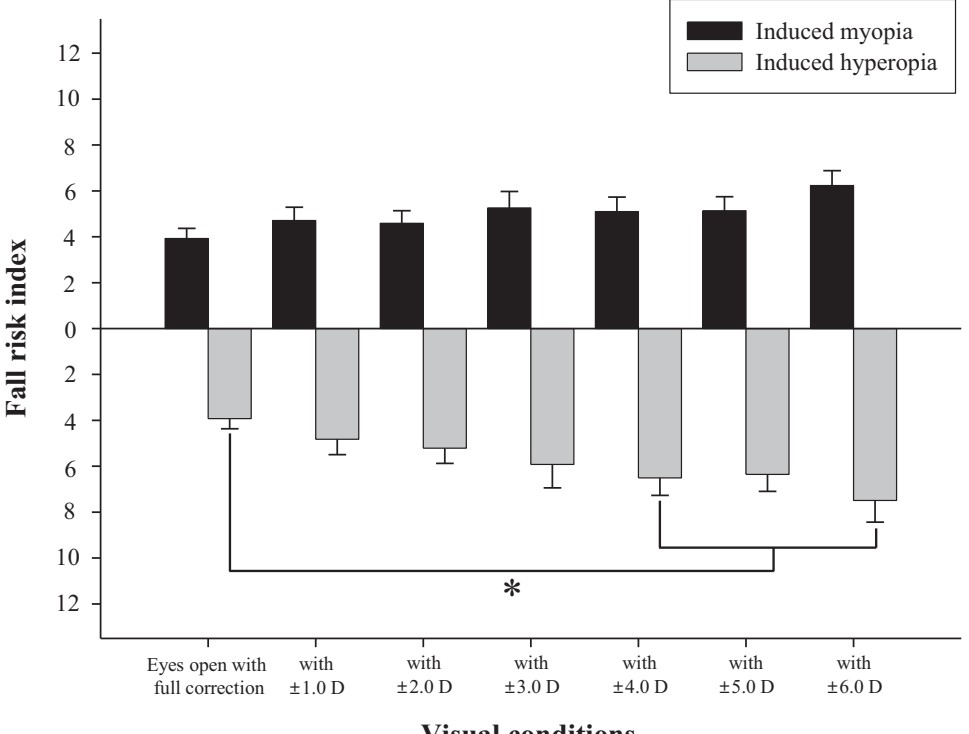

**Figure 4 Changes of fall risk index in hyperopic and myopic refractive errors induced by (±) spherical lenses compared to eyes-open with full correction condition.** *Significantly different from eyes-open with full correction condition according to repeated measures ANOVA. Error bars indicate the standard error (SE) of the mean.

showed that when the visual information was completely blocked, the sway of the static posture increased by 20–70% (*Lord, Clark & Webster, 1991*; *Isotalo et al., 2004*; *Henriksson et al., 1966*). As shown in Fig. 2, this study used Tetrax biofeedback system to reestablish the role and importance of visual information. It simultaneously confirmed the reliability of the instruments used in the study and the validity of measured values in that the fall risk noticeably increased in the eyes-closed compared to eyes-open with full correction.

Many previous studies have shown that refractive error clearly interferes with physical balance, but most of them focused on myopic refractive errors that lead to poor visual acuity (*Ivers et al., 1998*; *Klein et al., 2003*; *Edwards, 1946*; *Straube, Paulus & Brandt, 1990*). The main objective of this study was to investigate whether hyperopic refractive error was a major factor in increasing the risk of falls and to investigate whether the risk of falls was absolutely related to visual acuity. As shown in Fig. 4, there was no significant difference in the fall risk index with increase in myopia compared to eyes-open with full correction; however, the risk of falls was significantly increased compared to eyes-open with full correction from induced hyperopia to S −4.0 D. Interestingly, despite a drastic decline in decimal visual acuity when myopia was induced, there was no significant change in the fall risk index, whereas hyperopia was induced to S −4.0 D, which led to a significant increase in the fall risk index, and maintained the average visual acuity

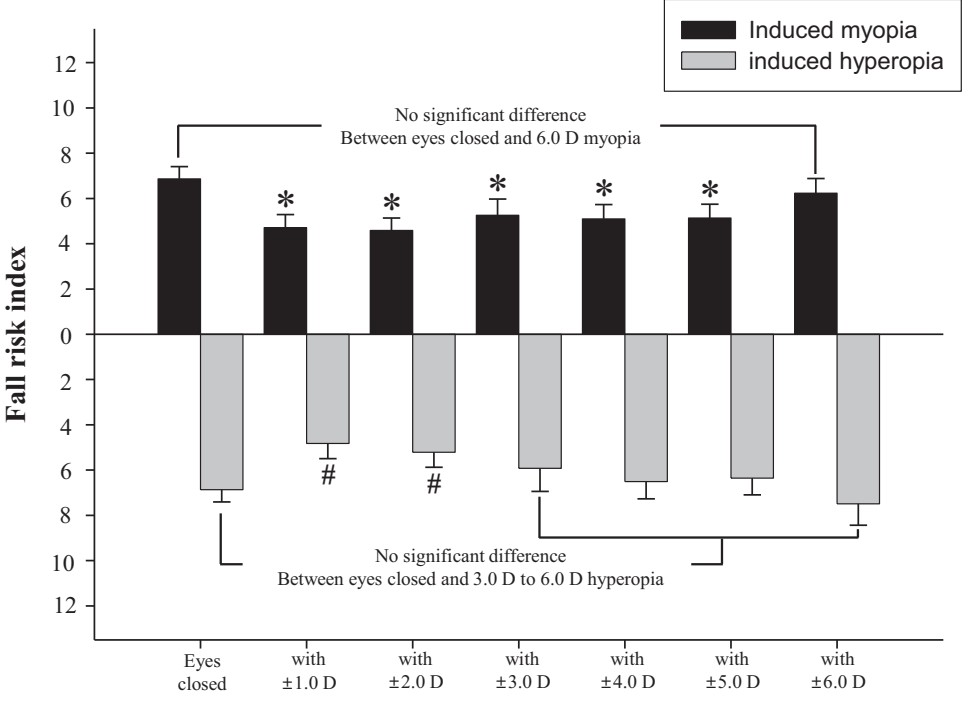

**Figure 5 Changes of fall risk index in hyperopic and myopic refractive errors induced by (±) spherical lenses compared to eyes-closed condition.** *Significantly different from eyes-closed condition in induced myopia according to repeated measures ANOVA. #Significantly different from eyes-closed condition in induced hyperopia according to repeated measures ANOVA. Error bars indicate the standard error (SE) of the mean.

**Table 1 Changes of sway power in frequencies bands derived from each subsystem according to an increase of induced myopia.**

| Lens power for myopia induction (diopter) | Sway power of each frequency band by Fourier transformation | | | |
|---|---|---|---|---|
| | Low | Low-to-medium | High-to-medium | High |
| 0.0 (Full correction) | 23.38 ± 10.53 | 9.26 ± 3.73 | 3.10±1.22 | 0.70±0.97 |
| +1.0 | 23.27 ± 14.86 | 9.13 ± 4.53 | 3.06 ± 1.49 | 0.70 ± 1.06 |
| +2.0 | 23.06 ± 13.83 | 8.89 ± 4.31 | 3.07 ± 1.41 | 0.73 ± 1.22 |
| +3.0 | 23.51 ± 15.69 | 9.28 ± 4.41 | 3.19 ± 1.36 | 0.72 ± 1.06 |
| +4.0 | 23.53 ± 12.61 | 9.64 ± 3.84 | 3.42 ± 1.31 | 0.70 ± 0.90 |
| +5.0 | 22.51 ± 16.95 | 9.30 ± 4.39 | 3.28 ± 1.35 | 0.73 ± 0.92 |
| +6.0 | 25.71 ± 18.61 | 9.81 ± 4.44 | 3.44 ± 1.56 | 0.76 ± 0.92 |
| $F/p$-value | 0.294/0.937 | 0.680/0.666 | 0.933/0.481 | 0.582/0.743 |

Notes:
Data are expressed as mean ± SD.
$n = 51$ (for each visual condition).

of 0.69 ± 0.32. In addition, we compared the risk of falls with eyes-closed (Fig. 5), which was S −3.0 D in hyperopia and S +6.0 D in myopia, and the visual acuity was 0.84 ± 0.23 and 0.05 ± 0.05, respectively. Hyperopia is definitely a type of refractive error, which

**Table 2 Changes in sway power of frequency bands derived from each subsystem according to an increase in induced hyperopia.**

| Lens power for hyperopia induction (diopter) | Sway power in each frequency band by Fourier transformation | | | |
|---|---|---|---|---|
| | Low | Low-to-medium | High-to-medium | High |
| 0.0 (Full correction) | 23.38 ± 10.53 | 9.26 ± 3.73[a] | 3.10 ± 1.22 | 0.70 ± 0.97 |
| −1.0 | 23.06 ± 12.60 | 9.35 ± 4.28[a] | 3.16 ± 1.13 | 0.65 ± 0.75 |
| −2.0 | 23.04 ± 15.97 | 9.49 ± 4.46[a] | 3.22 ± 1.23 | 0.71 ± 0.98 |
| −3.0 | 22.31 ± 14.72 | 10.10 ± 4.98[a] | 3.19 ± 1.40 | 0.76 ± 1.22 |
| −4.0 | 23.54 ± 15.45 | 10.58 ± 5.12[a] | 3.24 ± 1.31 | 0.81 ± 1.26 |
| −5.0 | 22.45 ± 10.75 | 10.28 ± 4.36[a] | 3.30 ± 1.21 | 0.79 ± 1.23 |
| −6.0 | 26.49 ± 20.21 | 11.31 ± 5.09[b] | 3.47 ± 1.26 | 0.90 ± 1.66 |
| $F/p$-value | 0.824/0.558 | 4.981/0.017* | 0.653/0.688 | 0.729/0.628 |

**Notes:**
Data are expressed as mean ± SD.
* $p < 0.05$: significant differences by repeated-measures ANOVA.
[a, b] subgroups by LSD (Least significant difference) post-hoc analysis.
$n = 51$ (for each visual condition).

shifts the focus behind the retina. However, unlike myopia, hyperopia results in additional optical power based on automatic focusing ability (accommodation), which enables image formation on the fovea as in emmetropia (*Benjamin, 2006*). Such hyperopia results in good visual acuity without optical correction in eyes with sufficient amplitude of accommodation. In general, poor visual acuity may be regarded as a major factor impeding postural stability; however, we suggest that poor visual acuity is not an absolute factor increasing the risk of falls. Factors that increase the risk of falls may differ depending on the type of refractive error and enhance the negative effect, especially in hyperopic refractive error rather than in myopia. This study supports our hypothesis that uncorrected hyperopia can be misinterpreted as emmetropia by maintaining appropriate visual acuity via accommodation, although this is clearly not the same as visual information derived from emmetropia.

We analyzed the changes in sway power across four frequency ranges derived from each subsystem using the Fourier transformation method provided by the Tetrax biofeedback system in order to identify the cause of increased risk of falls when hyperopic refractive error was induced. As a result of the study, the sway power showed a significant increase only in the low-to-medium frequency band associated with the peripheral vestibular system when the hyperopic refractive error increased, and the post-hoc analysis showed a significant difference in hyperopia induced to S −6.0 D compared to eyes-open with full correction (Table 2). In a related clinical study conducted using the Tetrax system, *Taguchi (1978)* and *Kollmitzer et al. (2000)* found that patients with pathologies involving the peripheral vestibular system showed a significant increase in sway power typically in the low-to-medium frequency band. Although hyperopic refractive error cannot be assessed as a serious pathological condition, elevated hyperopia of 6.0 D or higher showed a postural sway with characteristics similar to an abnormal shape seen in patients with vestibular disease. The parasympathetic nerve originating in the

Edinger–Westphal nucleus forms synapses at the ciliary ganglion, and innervates the sphincter pupillae and ciliary muscle predominantly. Contraction of the ciliary muscle contributes to the accommodation function by increasing the power of the crystalline lens (*Benjamin, 2006*). As shown in this study, induction of hyperopic refractive error in the subjects requires continuous efforts to form images at the fovea with sufficient accommodative capability. The resulting excessive involuntary parasympathetic nerve stimulation increasing the power of the crystalline lens may induce a temporary imbalance in the autonomic nervous system. *Takeda (2006)* explained that autonomic nerve system imbalance causes asymmetrical blood flow in the vertebral artery, which triggers asymmetrical activity of the vestibular nuclei or the inner ear resulting in dizziness. Despite the technical limitations involved in identifying a clear etiology in this study, the results of Fourier analysis based on the Tetrax system suggest that the imbalance in the autonomous nervous system due to excessive accommodation possibly induced a temporary disturbance in the vestibular system and increased the sway power in the low-to-medium frequency band.

*Jack et al. (1995)* found that 76% of patients hospitalized due to a fall accident had visual problems, and 79% of them recovered their visual acuity via re-correction of refractive error (40%) or cataract surgery (37%). Therefore, appropriate optical correction of refractive error should be perceived as an important factor in preventing frequent fall accidents among the elderly. In particular, in case of elderly with general hyperopic refractive changes (*Romín et al., 2015*) or patients with a low vision and high hyperopia (*Klimek et al., 2004*), optical correction of hyperopic refractive error is clinically significant in reducing fall risk despite the lack of any dramatic improvement in visual acuity. Treatment of uncorrected hyperopia by optometrists may prevent falling accidents in older people and requires close cooperation with professionals related to balance assessment and rehabilitation.

This study was intended to analyze the impact of pure refractive error on healthy young adults excluding physical factors affecting balance substantially. Although the study results reflected changes in the range of low-fall risk, the induced hyperopic refractive error increased the fall risk among individuals despite appropriate visual acuity. As this study results have limited validity and restricted to experimentally induced hyperopic refractive error, further studies are needed to provide a more realistic validation in subjects with real hyperopia and in patients with low vision.

## CONCLUSIONS

Fall risk increased significantly when hyperopic rather than myopic refractive error was induced, and visual acuity was not an absolute criterion for the evaluation of physical balance and fall risk. An excessive parasympathetic nerve stimulation resulting from a strong accommodation in a high hyperopic state of 6.0 D or above impedes the vestibular system, which contributes to posture adjustment. This study result confirmed that uncorrected hyperopia, despite adequate visual acuity based on effective accommodation system of the eyes, is definitely a type of refractive error, which increases the fall risk by impeding the postural adjustment.

In conclusion, the authors emphasize that corrected state of ametropia is more important than visual acuity as criteria for appropriate visual input underlying stable postural adjustment. Professionals working on postural assessment and rehabilitation need to be acquainted with this fact and are required to cooperate with optometrists or ophthalmologists.

### Funding
The authors received no funding for this work.

### Competing Interests
The authors declare that they have no competing interests.

### Author Contributions
- Byeong-Yeon Moon analyzed the data, conceived and designed the experiments, prepared figures and/or tables, authored or reviewed drafts of the paper, and approved the final draft.
- Jae Hyeok Choi performed the experiments, authored or reviewed drafts of the paper, and approved the final draft.
- Dong-Sik Yu analyzed the data, authored or reviewed drafts of the paper, and approved the final draft.
- Sang-Yeob Kim analyzed the data, conceived and designed the experiments, performed the experiments, prepared figures and/or tables, authored or reviewed drafts of the paper, and approved the final draft.

### Human Ethics
The following information was supplied relating to ethical approvals (i.e., approving body and any reference numbers):

This research complied with the tenets of the Declaration of Helsinki and was approved by the Institutional Review Board of Kangwon National University (KWNUIRB-2018-04-004-007).

### Data Availability
Raw data for data analyses and Figs. 2–5 and Tables 1 and 2 are available as a Supplemental File.

### Supplemental Information
Supplemental information for this article can be found online at http://dx.doi.org/10.7717/peerj.8329#supplemental-information.

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
