# Peer review of "Effect of induced hyperopia on fall risk and Fourier transformation of postural sway"

_PeerJ, doi:10.7717/peerj.8329_

## Round 0.1 · original submission · Minor Revisions

The authors have addressed an interesting topic. Some minor revisions should be considered.

Reviewer 1 ·

Basic reporting

The manuscript was written clearly. The results of abstract didn't clearly mention the results (difference and p-values etc.)

Experimental design

I am not expert on Hyperopia, but whether given spherical lens to the same person with 10-minute break will affect the results? Or will need reference on this.

Validity of the findings

The statistical methods used in the manuscript were correct and interpreted correctly. I would suggest adding p-values after multiple testing correction to make sure the significant findings still exist after multiple tests correction.

Additional comments

The discussion parts were clearly stated.

Reviewer 2 ·

Basic reporting

Reviewer read the manuscript “Effect of induced hyperopia on fall risk and Fourier transformation of postural sway” by Sang-Yeob Kim and co-workers with interest. This is a study the effect of refractive errors on their fall risk. The results indicate that fall risk index is more sensitively affected by uncorrected hyperopia than uncorrected myopia.
The topic is quite novel and results are very exciting findings.

Experimental design

Abstract
- If significant, p-values need to be provided.

Introduction
- It will be help to strengthen the purpose of this paper, if discuss how increase fall risk linked to poorer quality of life.

Subject and methods
- It would be better if you added the issue of the reliability of the measuring device.
- Line 134-135. What was paired t-test for, and what was repeated AVONA for? Additional explanation is needed.

Results
- Figures: what do the error bars mean? Additional explanation is needed in foot notes below the figures.

Discussion
- Line 228-230: In explaining the limitations of this study, you mentioned the problem caused by the difference between the induced hyperopia and the actual hyperopia… Do you think that the same results can be achieved between the induced hyperopia and the actual hyperopia?

- Line 195. 200, 215. ‘low-to-medium’ it is need to correct the letter style as ‘Time New Roman’
- Line 196. 199. ‘Peripheral’ it is need to correct the letter style as ‘Time New Roman’

References
- References do not have serial numbers, but there are reference numbers in the text.

Validity of the findings

The experiments are performed on satisfactory number of subjects and the manuscript is written in a good style. Reviewer still has some recommendations which would improve the quality of the work.

Additional comments

- This study is about the fall index in induced myopia or induced hyperopia. However, I think the fall risk index may be increased by the prism effect of the added lens, or the narrowing of the visual field on the trial frame. Do you consider this issues?
- This study is saying that the fall risk index rises because of the accommodative action in uncorrected hyperopia. However, people of higher age have little accommodative amplitude, so they cannot use accommodation even with uncorrected hyperopia. Doesn't the fall risk index rise for older people because of uncorrected hyperopia?

Reviewer 3 ·

Basic reporting

Your introduction needs more detail. I suggest that you improve the description at lines 64- 65 to provide more research results about proportion of hyperopia in Korea and why it was rarely studied before compared to myopia.

Experimental design

In Materials & Methods section, subjects are in 20’s and they averagely have myopic and astigmatic refractive error. The participants were accustomed to myopic condition, so it must be needed some time to adjust hyperopic condition, because it is not common that someone gets hyperopic changes all of sudden. Please explain did you have some time to adapt themselves to new circumstances.

Validity of the findings

Induced hyperopia has other problems such as limited vision and magnifying images. The authors should have mentioned in discussion section.

Additional comments

The authors have done some statistical testing which is good, however the results could be presented more clearly. The statistical testing results are presented in figure 5, however normally when reporting on ANOVA tests the degrees of freedom and F value should be reported as well. See http://www.yorku.ca/mack/RN-HowToReportAnFStatistic.html for an example of how to report ANOVA values. The authors should also report on the numerical scores resulting from post-hoc analysis and what kind of post-hoc analysis did you use.

---

## Round 0.2 · accepted · Accept

The manuscript has been improved following the reviewers suggestions.